# Generation and Evaluation of an Efficient Femtosecond Green Laser

**DOI:** 10.3390/s24165240

**Published:** 2024-08-13

**Authors:** Mingyang Teng, Xianghao Meng

**Affiliations:** School of Physics and Optoelectronic Engineering, Beijing University of Technology, Beijing 100124, China; 13121819621@163.com

**Keywords:** femtosecond, laser, power stability, pulse duration measurement, 71.20.Eh, 42.55.Wd, 42.65.Ky

## Abstract

We demonstrate femtosecond ultra-stable green laser generation by an ytterbium-doped polarization-maintaining fiber laser with a 2.4 mm long lithium triborate (LBO) crystal. We generated 5.6 W of femtosecond green light at 520 nm for a fundamental power of 12 W, which corresponds to a conversion efficiency of 46.7%. The fiber chirped-pulse amplifier, which has an environmentally immune front end, delivered 170 fs pulses at a 75 MHz repetition rate centered at 1040 nm. According to the dispersion of the optical material in a double-frequency setup, the introduced dispersion had a negligible effect for the green laser, and the pulse duration of the generated green laser was calculated to be 171 fs, resulting in an excellent power stability, with fluctuation as low as 0.16% of the generated green light. This system could be of great interest in ultrafast optical and photobiology research.

## 1. Introduction

Ultrashort pulsed green light is attractive in the industrial and scientific fields and well-suited in enabling optical parametric oscillators [1], photobiology imaging [2] and ultrafast optical research [3,4,5]. Such pulses can be achieved by the efficient frequency conversion of a Yb^3+^-based laser through nonlinear crystals like potassium dihydrogen phosphate (KDP) [6], barium metaborate (BBO) [7] and lithium triborate (LBO) [1]. However, due to thermal instability of power scaling of solid-state lasers, the power stability and beam quality deteriorates rapidly. Compared to their solid-state counterpart, frequency doubling in Yb^3+^-fiber lasers has been proven attractive with excellent power stability, outstanding thermo-optical properties, compact structure and good environmental immunity. It achieved high average power [8], short pulse duration [9], and near-diffraction-limited beam quality [10] at a central wavelength of 1 μm. By employing the principle of fiber chirped-pulse amplification (FCPA), Rothhardt J. et al. achieved an average power up to hundreds of watts [11] with a femtosecond pulse duration. The second-harmonic (SH) pulse energy of such laser pulses also achieved μJ level. But the introduction of massive free-space elements makes it hard to increase the power and extend its application. On the other hand, ultrashort green-pulse generation requires “thin” nonlinear crystals to meet the phase-matching condition of a broad spectral range, which in turn limits the SHG efficiency. By optimizing condition and crystal length, N. A. Chaitanya et al. generated 2.25 W green radiation of pulse duration of 176 fs, with single-pass conversion efficiency of 46.5%. The relatively small SH power lay on the low output of front-end laser [12]. In this paper, we created a polarization-maintaining fiber chirped-pulse amplification system, extending the IR output to 15 W. By using a pair of diffraction grating compressors, 170 fs laser pulses centered at 1040 nm were exported. An average power of 5.6 W output with a central wavelength of 520 nm was obtained at 12 W s-polarized Infrared (IR) with the laser focusing on LBO, which corresponds to a conversion frequency of 46.7%; the estimated SH pulse duration was 171 fs. With an excellent output fluctuation as low as 0.16% of the generated SH pulses, this high-power ultrashort-pulse green source shows high potential in the industrial and scientific fields.

## 2. Experimental Setup and Discussion

The schematic of the experimental setup is illustrated in Figure 1. The front-end high-power ultrashort-pulse Yb-fiber laser consists of an oscillator, a pulse stretcher made up of a 20 m long single-mode fiber (HI1060), a core-pumped single-mode gain-fiber pre-amplifying stage, two air-cooled double-clad gain-fiber main amplifying stages and a grating pair-based pulse compressor. The single-pass second harmonic generation setup was built after this laser.

The seed pulses were provided by a dispersion-managed Yb-doped nonlinear polarization evolution (NPE) fiber oscillator with a spectral bandwidth (FWHM) of 38 nm centered at 1048 nm and 1.3 ps chirped pulse duration at a 75 MHz repetition rate. This system consists of fiber and free-space sections. The fiber section consists of a WDM, two segments of single-mode fiber, two collimators, a 30:70 OC and a 25 cm long SC-YDF with an absorption coefficient of 473 dB/m at 915 nm. The free-space section consists of an HWP, two QWP, a polarization-dependent isolator, a 1000 lines/mm transmission grating pair, two high-reflection mirrors and a PBS. The 976 nm laser diode is coupled into a cavity via WDM as a pump source. The isolator ensures the laser unidirectional propagation in the ring cavity. The intra-cavity grating pair can introduce a certain amount of negative group delay dispersion (GDD). By carefully adjusting the angle of the wave plates and the distance of the grating pair, self-starting mode-locked operation can be built through nonlinear polarization evolution. The output was taken directly from the 30:70 OC, and 20 mW output power could be obtained. The output pulse train is shown in Figure 2. 

Figure 3a,b shows the spectrum (980–1120 nm, resolution, 0.2 nm, AQ6370D, YOKOGAWA) and the RF spectrum (BW: 10 kHz, E4402B, Agilent, CA, USA). The spectrum of oscillator and compressor are depicted in brown and blue curves, and their central wavelength and FWHM were 1048 nm (38 nm) and 1040 nm (10.1 nm). It can be seen clearly that narrowing of the output spectrum and shifting of the central wavelength to 1040 nm occurred after power amplification, which could be attributed to the strong gain-narrowing effect of Yb-doped fiber and the differences in band width of each device. The signal-to-noise ratio (SNR) of the fundamental frequency from the oscillator was 80 dB, which indicated robust mode-locking and low simultaneous noise ejection.

In order to eliminate the nonlinear effects such as self-phase modulation, self-focusing and stimulated Raman scattering induced by the high peak power density during three-stage gain-fiber amplification, the pulse duration of the oscillator should be stretched to a certain amount. After propagating through a 20 m long single-mode fiber to introduce a normal GDD of 0.48 ps^2^, the injecting pulse duration of 1.3 ps was stretched to 20 ps. Considering the output stretched pulse was elliptically polarized, an inline polarizer was used to convert it to a linearly polarized pulse with a polarization extinction ratio (PER) of 22.1 dB. After that, the seed pulses were injected into three cascade stage amplifiers to obtain a high average power. The isolators employed between the adjacent two-stage fiber-amplifying stages were used to keep the front end immune from the anti-direction laser. The first power amplifying stage consisted of polarization-maintaining single-cladding Yb-doped fiber (PM-Yb401, Coractive, absorption at 976 nm of 150 dB/m), a WDM and a pump diode centered at 976 nm. Then, a 3 m long PM-SC-YDF was employed to provide sufficient gain for a small signal amplification. The output power was 300 mW, with 600 mW pump power, and the spectral bandwidth was 33 nm centered at 1043 nm. After that, a small-core double-cladding fiber amplifier was introduced to obtain higher output power. During the DCF amplification, the pump beam delivered by a fiber-coupled laser diode was confined to the inner cladding, whereas the signal beam propagated in the core area; this characteristic improved the amplification efficiency due to the better absorption of the pump laser that could interact with Yb^3+^ ions in a larger area. The 5 W pump laser was coupled into the cladding area of the fiber by a beam combiner, for WDM could not hold such high average power. A 3 m long polarization-maintaining double-cladding Yb-doped fiber (PLMA-YDF-10/125, Nufern, absorption at 976 nm of 4.8 dB) could export 2.63 W of signal power. In order to eliminate the high-order modes lasing in the double-cladding gain fiber, we rolled this fiber in an aluminum post with a curvature radius of 10 cm, which could provide bending loss for the high-order modes while maintaining a low loss for the fundamental mode. The pre-amplified pulses were finally launched into the 2 m long highly doped polarization-maintaining double-cladding Yb-doped fiber (PLMA-YDF-30/250, Nufern, absorption at 976 nm of 6 dB) with a core diameter of 30 μm. A 30 W laser diode was used as a pump source. The variations of the amplified uncompressed and compressed output power as a function of the pump power are shown in Figure 4.

The maximum output power was 15 W (measured by 3 sigma, Laser Power Energy Meter, COHERENT) at the pump power of 30.2 W. The corresponding optical efficiency of the three amplifiers was 50%, 52.6% and 49.7%. With a compressing efficiency of 80%, a 12 W final output power could be obtained. The output from the main amplifier was then launched into free space through a high-power gain fiber-pigtailed collimator with a maximum damage threshold of 20 W average power, which limited a further increase in output power. In order to reduce the back reflection and prevent end-facet damage, the end of the collimator was high-transmission coated at a wide spectral range from 1020 nm to 1100 nm. A pair of diffraction gratings (1000 lines/mm, Light Smyth) in a double-pass configuration was employed to compress the output pulses from the amplifier. The maximum diffracted efficiency was 95% when they were mounted at a certain angle (31 degrees for 1030 nm). By careful adjusting the angle and the distances between the gratings to introduce an appropriate amount of GDD of −5.94 × 10^5^ fs^2^, the optimum compressed pulses duration was obtained with a small pedestal in autocorrelation trace (pulseLink, A.P.E), which is shown in Figure 5. 

It can be seen in Figure 3a that the spectrum of the final output pulses (blue curve) had a bandwidth of 10.1 nm centered at 1040 nm, corresponding to a Fourier transform-limited pulse duration of about 100 fs. But the existence of a 5 nm deep gap in the 3 dB spectra could make it difficult to further compress of output pulses to transform the limited pulse duration. The trace shows that the incompressibility was attributed to the existence of third-order dispersion (TOD) introduced by the single-mode fiber stretcher and grating-pair compressor, which formed the pedestal and accumulated nonlinear phase during amplification. The 170 fs optimum pulse duration was obtained at the average output power of 15 W directly from the amplifier and fluctuated slightly during power scaling. It is noteworthy that the optimum pulse duration of this fiber chirped-pulse amplification system also depends on other parameters (e. g., the input pulse duration from the oscillator, which is constant once the oscillator is assembled, the fiber length in the stretcher, the pump power on each stage, etc.), as this fiber chirped-pulse amplification system is polarization-maintaining. However, after power amplification and compression, the PER of the signal pulse was reduced from 22.1 dB to 14 dB. A high-power half-wave plate was introduced to adjust the polarization state of the IR pulses, then the output which had a beam radius of 3 mm was focused on a 2.4 mm LBO crystal by a high-transmission-coated focal lens at 1 μm with a radius of curvature of 50 mm. Although potassium titanyl phosphate (KTP) and bismuth borate (BiBO) own even larger nonlinear coefficients, they suffer from gray tracking [13] and color center formation [14] at high peak power density, respectively. LBO is widely used in frequency conversion with its high damage threshold (18.9 GW/cm^2^) and effective nonlinear coefficient of 1.05 pm/V. In this experiment, a 5 × 5 × 2.4 mm^3^ LBO piece cut at Φ = 13.6 degrees and θ = 90 degrees was used for type I (e + e→o) phase matching in the XZ plane, whose facets were also anti-reflection-coated (1040/520 nm) for both the green and the IR wavelengths to avoid excess losses at the interfaces. By slightly rotating the HWP and BBO mount to meet the collinear type I phase-matching condition, the maximum output power of 5.6 W was obtained at an injection pump power of 12 W, which corresponds to a conversion efficiency of 46.7%. Then, we used a dichromic mirror (HR@520 nm; HT@1040 nm) to split the second-harmonic green laser from the fundamental frequency IR laser. An f = 50 mm focal lens was used to collimate the reflected green laser and directly obtain the collimated laser beam with 1.3 and 1.4 beam quality factors M^2^ (M2-200S, Spiricon, Jerusalem, Israel) in the horizontal and vertical directions, respectively, as shown in Figure 6.

The measured power fluctuation of the SH output is also shown in Figure 6, with the whole setup working in air-cooled, free-running conditions. The approximately 0.16% root mean square (RMS) over 4 h strongly suggests the high performance of this setup. The output spectrum (480–555 nm, resolution, 1 nm, Avantes) is shown in Figure 7.

The FWHM of the spectra was 4 nm, with central wavelength of 520 nm, which could sustain a transform-limited pulse duration of 71 fs. However, because the CPA system TOD affects the IR pulse profile, the SH pulse duration would be wider than that. Considering the GDD introduced by the SH setup (detailed statistics are shown in Table 1) and assuming that the generated SH laser had a pulse duration of 170 fs as the IR pulses, the accumulated GDD of 1103 fs^2^ would extend the SH pulse duration to 171 fs. This shows that the introduced GDD of the SH setup has a negligible effect on SH pulse duration.

## 3. Conclusions

In conclusion, we reported a highly stable fiber-based chirped-pulse amplification system generating a maximum output power of 12 W with a pulse duration of 170 fs centered at 1040 nm. Using it as a pump source, harmonic generation of 520 nm based on a 2.4 mm long lithium triborate (LBO) crystal could be obtained. The maximum power was 5.6 W at a 12 W s-polarized pulse pumping, which corresponds to a conversion efficiency of 46.7%, and the power fluctuation was as low as 0.16%. The estimated pulse duration of the SH output green laser is 171 fs.

## Figures and Tables

**Figure 1 sensors-24-05240-f001:**
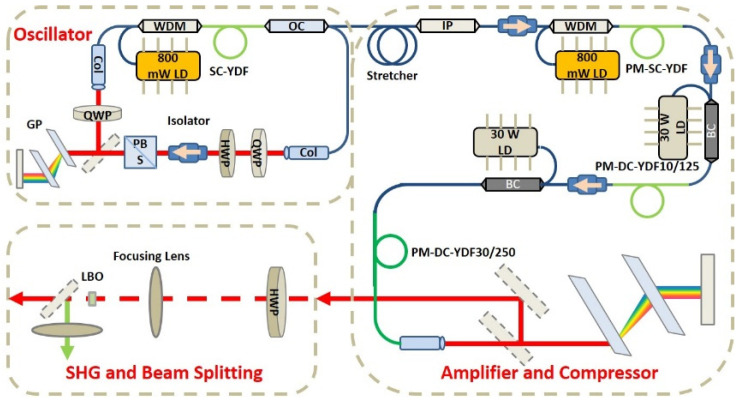
Schematic diagram of the high-power femtosecond Yb-fiber laser and its SHG configuration. WDM: wavelength division multiplexer; LD: laser diode; SC-YDF: single-clad Yb-doped fiber; OC: output coupler; Col: collimator; QWP (HWP): quarter–(half)-wave plate; PBS: polarization beam splitter; GP: grating pair; IP: inline polarizer; PM-SC-YDF: polarization-maintaining single-cladding Yb-doped fiber; BC: beam combiner; PM-DC-YDF: polarization-maintaining double-cladding Yb-doped fiber; LBO: lithium triborate.

**Figure 2 sensors-24-05240-f002:**
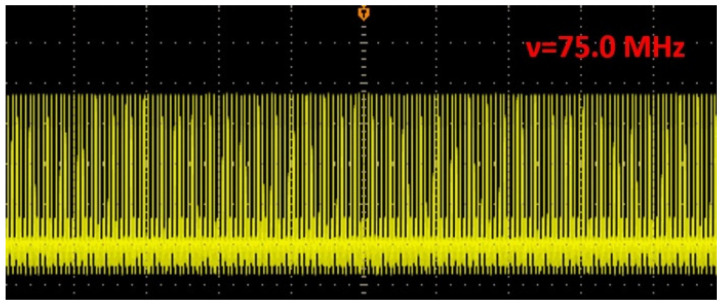
The mode-locked pulse train of the oscillator with a repetition rate of 75.0 MHz.

**Figure 3 sensors-24-05240-f003:**
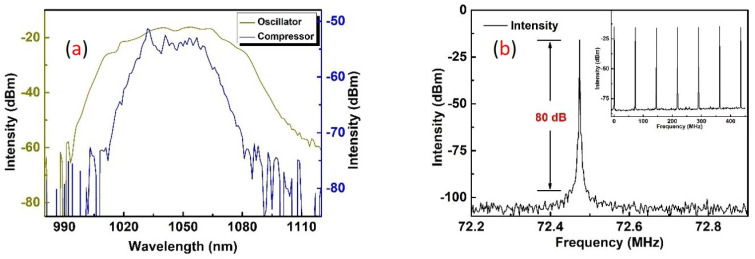
(**a**) The spectra of oscillator (brown) and compressor (blue); (**b**) the RF spectrum of the oscillator (insert: RF spectrum from −10 MHz to 460 MHz).

**Figure 4 sensors-24-05240-f004:**
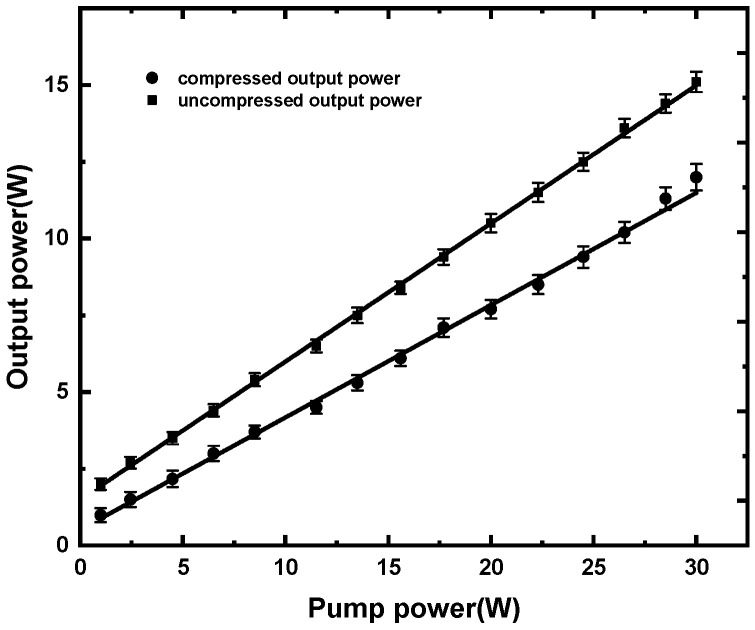
Output power of uncompressed and compressed pulses as a function of the pump power.

**Figure 5 sensors-24-05240-f005:**
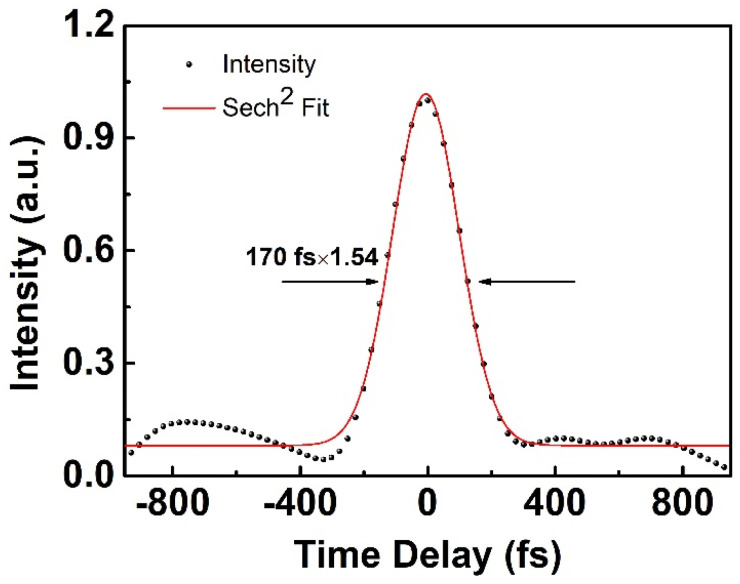
Normalized autocorrelation trace of compressed pulses. Dots: experimental results; red curve: Sech^2^ fitting of the experimental results.

**Figure 6 sensors-24-05240-f006:**
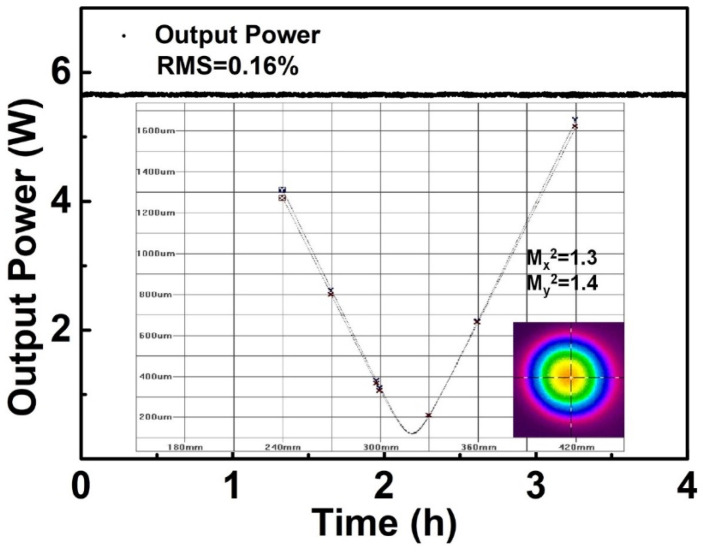
Measured power stability, M^2^ value and beam profile of the SH output.

**Figure 7 sensors-24-05240-f007:**
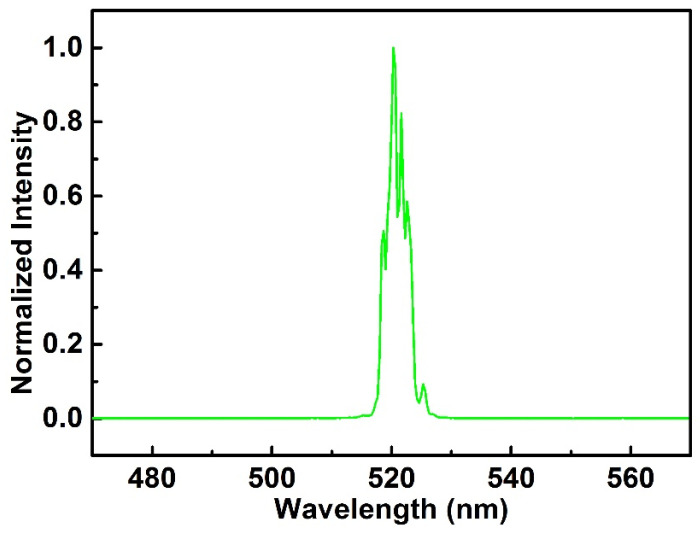
The spectrum of output SH pulses.

**Table 1 sensors-24-05240-t001:** Total GDD of SH setup.

	Half-Wave Plate	f = 50 mm Focus Lens	LBO Crystal	f = 50 mm Collimate Lens
Texture	Silica	Fused silica	LBO	Fused silica
GVD(@520nm)	67.8 fs2/mm	67.8 fs2/mm	86.8 fs2/mm	67.8 fs2/mm
Thickness	3 mm	5.7 mm	2.4 mm	5.8 mm
Total GDD	1103 fs2			

## Data Availability

The data presented in this study are available from the corresponding author upon reasonable request.

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
