# Peer review of "Generation and Evaluation of an Efficient Femtosecond Green Laser"

_sensors, 2024, doi:10.3390/s24165240_

Round 1

Reviewer 1 Report

Comments and Suggestions for Authors

The manuscript (ID: sensors-3114858, title: Generation and measurement of efficient femtosecond green laser) demonstrated a highly efficient femtosecond green laser, include generation and its optical sensor measurement. Based on home-made all-Yb-fiber CPA amplifier with output power of 12 W, using LBO crystal, author got 5.6 W of green fs laser, which corresponds the conversion efficiency of 46.7%, more important feature is the high stability in the natural environment, which is benefit for optical sensors applications such as photobiology sensor and ultrafast physical research. Based on above, the results are good and the paper is well organized and written, the manuscript meets the requirement of SENSORS. I agree its publication on SENSORS.

However, I still have a few questions and suggestions for the authors.
(1) I suggest that “Author should add the pulse duration of 171 fs when description of green fs laser in the abstract paragraph” .

(2) As the length of the LBO crystal, why chose the length of 2mm, compared to other length?

(3) Suggestions on grammar, for the description of the experiments, the author should use the past tense instead of present tense.

Author Response

Thank the reviewer’s comments, we have carefully read and considered the comments and suggestions.  We have revised the manuscript accordingly and all the questions have been well addressed in new version. We do hope your favorable reconsideration of the revised manuscript for publication.

Our responses are as followings:

Comment 1: I suggest that “Author should add the pulse duration of 171 fs when description of green fs laser in the abstract paragraph”

Response 1: Thanks, we agree it. We already revised it and add the description of pulse duration of 171 fs for the green fs laser in the abstract paragraph, and the revision is marked in red.

Comment 2: As the length of the LBO crystal, why chose the length of 2mm, compared to other length?

Response 2: The double-frequency efficiency depends on the crystal length, but not the longer the better, but also depend on the walk-off, the larger walk-off angle, the lower the efficiency, so according to the balance between the femtosecond pulse duration and the walk-off effect, the best crystal length for the 170 fs laser pulse is calculated as 2.4 mm. We also compared the efficiency of different thickness crystal in the experiment and found that the efficiency of 2.4 mm was highest, which has 46.7%. As comparison of other publications, the efficiency is also very high.

We also re-calculated the total dispersion (GDD) of all optical components in SH setup, and found total GDD is 1103 fs2 , instead of 1185 fs2, so we corrected the errors, and marked in red.

Comment 3: Suggestions on grammar, for the description of the experiments, the author should use the past tense instead of present tense.

Response 3: Thanks, good suggestions. We revised the manuscript according to the suggestion, the revision is marked in red.

Reviewer 2 Report

Comments and Suggestions for Authors

The authors designed and constructed a highly stable fiber-based chirped pulse amplification system,and generate a maximum output power of 12 W with a pulse width of 170 fs centered at 1040 nm. Using it as a pump source, with type I phase matching, obtain the maximum power of 5.6 W at 12 W S-polarized pulses pumping, corresponding to a conversion efficiency of 46.7%, and the estimated pulse width of SH output of 171 fs. These results have significant research value for 520nm ultrafine green laser. However, there exist a few questions in the manuscript, and after addressing these points, the manuscript would be suitable for publication.

1. How about the polarization characteristics of the output 520nm laser?

2.There are some details in the manuscript that should be noted. For example, "*" should be replaced with "×". For example,“A 5*5*2.4 mm3 LBO……”in line 176; “……of GDD of -5.94*105 fs2……” in line 147,and so on.

3. “……50%, 52.6 and 49.7%......” should be “……50%, 52.6% and 49.7%......” in line 136.

Comments on the Quality of English Language

4.  “a ytterbium-doped polarization maintaining fiber laser……” should be “an ytterbium-doped polarization maintaining fiber laser……”in line 7 of Abstract. “……in a aluminum post……” should be “……in an aluminum post……”in line 123.

Author Response

Thank! We have carefully read and considered the comments and suggestions.  We have revised the manuscript accordingly and all the questions have been well addressed. We hope your favorable reconsideration of the revised manuscript for publication.

Comment 1: How about the polarization characteristics of the output 520nm laser?

Response 1: Thanks. Due to all fibers in the laser amplifier is polarization-maintaining, and double transmission grating in compressor also require linear polarization, the output fs laser from the compressor is S- polarized. The double frequency LBO crystal is designed as type I phase-matching, so the polarization of 520 nm green laser is P-polarized.

Comment 2: There are some details in the manuscript that should be noted. For example, "*" should be replaced with "×". For example,“A 5*5*2.4 mm3 LBO……”in line 176; “……of GDD of -5.94*105 fs2……” in line 147,and so on.

Response 2: Thanks for pointing out the errors and suggestions. We revised the manuscript accordingly, and the revisions are marked in red.

Comment 3: “……50%, 52.6 and 49.7%......” should be “……50%, 52.6% and 49.7%......” in line 136

Response 3: Thanks. We modified the error, and marked it in red.

Comment 4: “a ytterbium-doped polarization maintaining fiber laser……” should be “an ytterbium-doped polarization maintaining fiber laser……”in line 7 of Abstract. “……in a aluminum post……” should be “……in an aluminum post……”in line 123.

Response 4: Thanks. We modified it and marked it in red.

Reviewer 3 Report

Comments and Suggestions for Authors

In this manuscript, the authors report a demonstration of highly efficient femtosecond green laser based on ytterbium-doped polarization maintaining fiber laser with LBO crystal, and got 5.6 W of fs green laser, which conversion efficiency is 46.7%. The output power stability is 0.16%, which is very good under normal circumstances.

Many optical sensors need stable green fs laser, and it is important for ultrafast optical and photobiology research, ultrafast photoemission electron microscopy, and ultrafast beam profile and photodiode sensor. In my opinion, the ultra-stable green fs laser is very important light source for optical sensor, the content of the manuscript meets the requirements of SENSORS journal, the manuscript can be accepted for publication with some grammatical and word corrections, for example, in Conclusion paragraph, the “ Corresponding” should be “ Which corresponds ”. Furthermore, the following related publications about optical sensors based on solid-state laser should be cited in this paper. [Room-temperature Fe:ZnSe laser tunable in the spectral range of 3.7–5.3 μm applied for intracavity absorption spectroscopy of CO2 isotopes, CO and N2O], [Ultra-highly sensitive dual gases detection based on photoacoustic spectroscopy by exploiting a long-wave, high-power, wide-tunable, single-longitudinal-mode solid-state laser].

Author Response

Thank! We have carefully read and considered the comments and suggestions.  We have revised the manuscript accordingly and all the questions have been well addressed. We hope that our revised manuscript is suitable for publication.

Comment 1: in Conclusion paragraph, the “ Corresponding” should be “ Which corresponds ”

Response 1: Thanks for suggestion, we already revised it and marked in red.

Comment 2: Furthermore, the following related publications about optical sensors based on solid-state laser should be cited in this paper. [Room-temperature Fe:ZnSe laser tunable in the spectral range of 3.7–5.3 μm applied for intracavity absorption spectroscopy of CO2 isotopes, CO and N2O], [Ultra-highly sensitive dual gases detection based on photoacoustic spectroscopy by exploiting a long-wave, high-power, wide-tunable, single-longitudinal-mode solid-state laser].

Response 2: Thanks for suggestion, we added the two related publications in the references, and marked in red.

Reviewer 4 Report

Comments and Suggestions for Authors

Author Response

Thank for positive consideration. We have carefully considered the comments and
suggestions, and have revised the manuscript accordingly and all the questions have
been well addressed.
Comment 1: Line 33, amplification (FCPA), Rothhardt J et al have achieved average
power up to
amplification (FCPA), Rothhardt J et al. have achieved average power up
to

Response 1: Thanks. We have modified it and marked it in red.
Comment 2: Line 34, dreds watts [9] with femtosecond pulse duration. The SH
pulse energy of such
““reds watts [9] with femtosecond pulse duration. The second
harmonics (SH) pulse energy of such
, the first abbreviation that appears should have
a full description.
Response 2: Thanks for the suggestion. We have modified the errors, and marked
it in red.
Comment 3: Line 129, “uncompressed/compressed output power as functions of
the pump power is shown in Fig. 4”, please add the uncertainty on the measured data.
Response 3: Thanks. We have modified the Fig.4 and add the uncertainty of data.
